# Chemical Composition of *Reichardia tingitana* Methanolic Extract and Its Potential Antioxidant, Antimicrobial, Cytotoxic and Larvicidal Activity

**DOI:** 10.3390/plants11152028

**Published:** 2022-08-04

**Authors:** Salama A. Salama, Zarraq E. AL-Faifi, Yasser A. El-Amier

**Affiliations:** 1Biology Department, Faculty of Science, Jazan University, P.O. Box 114, Jazan 45142, Saudi Arabia; 2Zoology Department, Faculty of Science, Damanhur University, Damanhour 22511, Egypt; 3Center for Environmental Research and Studies, Jazan University, P.O. Box 2097, Jazan 42145, Saudi Arabia; 4Department of Botany, Faculty of Science, Mansoura University, Mansoura 35516, Egypt

**Keywords:** *R. tingitana*, GC-MS, chemical components, biological activity, tumor cells, *Aedes aegypti*

## Abstract

The biggest challenges are locating effective, reasonably priced, and eco-friendly compounds to treat diseases caused by insects and microbes. The aim of this study was to employ GC-MS to assess the biological potency and chemical composition of the aerial parts of *Reichardia tingitana* (L.) Roth. Using this technique, 17 components were interpreted from the extracted plant, accounting for around 100% of total volatile compounds. Commonly, 6,10,14-trimethylpentadecan-2-one (21.98%) and methyl oleate (27.26%) were positioned as the major components, which were ascertained after 19.25, and 23.34 min, respectively. The major components were classified as hydrocarbons (23.82%), fatty acids, esters of fatty acids (57.46%), steroids (17.26%), and terpenes (1.48%). The DPPH antioxidant activity of the *R. tingitana* extracted components revealed that the shoot extract is the most powerful, with an IC_50_ value of 30.77 mg L^−1^ and a radical scavenging activity percentage of 71.91%. According to the current result, methanolic extract of *R. tingitana* had the maximum zone of inhibition against *Salmonella typhimurium* and *Bacillus cereus* (25.71 ± 1.63 and 24.42 ± 0.81 mm, respectively), while *Clostridium tetani* and *Staphylococcus xylosus* were the main resistant species. In addition, the 50% methanol crude shoot extract of *R. tingitana* showed greater potential anticancer activity with high cytotoxicity for two tumor cells HepG-2 and PC3 cells (IC_50_ = 29.977 and 40.479 µg mL^−1^, respectively) and noncytotoxic activity for WI-38 normal cells (IC_50_ = >100 µg mL^−1^). The MeOH extract of plant sample was more effective against *Aedes aegypti* larvae with LC_50_ of extract being 46.85, 35.75, and 29.38 mg L^−1^, whereas the LC_90_ is 82.66, 63.82, and 53.30 mg L^−1^ for the various time periods of 24, 48, and 72 h, respectively. *R. tingitana* is a possible biologically active plant. Future study will include pure chemical isolation and individual component bioactivity evaluation.

## 1. Introduction

Wild plants are part of nature’s biodiversity. They are rich in various bioactive molecules and nutrients, and are a great source of food for humans and agroindustry [1]. Since far before recorded history, wild plants have played a significant role in the suppression of dietary and pathogen-related ailments of native people. In most civilizations, particularly in developing countries, wild plants provide enormous direct potential benefits by serving as a major source of sold commodities. In European nations, around 1300 medicinal plants are utilized, with 90% of them being wild [2]. These discreet pharmaceuticals obtained from natural sources have less negative side effects than those obtained from synthetic sources [3].

Around 2080–2094 species of seed plants and vascular cryptogams are found in Egypt’s flora [4,5]. Asteraceae family is one of the biggest flowering plant families, with a global distribution. Although the family includes a substantial number of shrubs, vines, and trees, the majority of the members are herbaceous. The Asteraceae family has 98 genera and 228 species in Egypt’s flora [4,6]. Many biological actions have been identified for members of the Asteraceae family, including antibacterial, antifungal, antidiabetic, antihelminthic, immunostimulatory, and anticancer properties [7,8,9]. Only two *Reichardia* species, *R. picroides* (L.) Roth and *R. tingitana* (L.) Roth, were recorded in Egypt’s flora that grows abundantly in Egypt’s coastal and inland deserts. *R. tingitana* (False sowthistle) is a glabrous erect annual herb that is native to the Mediterranean region and western Asia. The stem branches from the base, the plant grows up to 40 cm tall with the taproot, and it blooms from March to May, with characteristic yellow capitula terminal, solitary or paired on swollen stalks [6]. This genus has eight species that are found across the Mediterranean region, with some species being found in portions of India and Africa [10].

Recent studies have focused on the identification of active chemical components, as well as the biological applications of extracted plants [11,12]. Tumors, hepatic illnesses, poor digestion, constipation, calculi, heart issues, pains, spleen disorders, and hiccough are just a few of the afflictions that the whole plant can help with [13,14]. Several viral diseases, such as dengue fever, chikungunya, Zika fever, Mayaro, and yellow fever viruses, as well as other disease agents, are all conveyed by the mosquito *A. aegypti*. Continuous use of synthetic pesticides leads to vector resistance, biological multiplication of hazardous compounds via the food chain, and negative consequences for environmental quality and nontarget creatures, including human health [15,16].

The chemistry of *R. tingitana* extract was reported and it showed antimicrobial, bioherbicide, and cytotoxic activities [9,17]. Constituents, such as phenolics, tannins, flavonoids, coumarins, phytosterols, lactones, and essential oils, have been isolated from aboveground biomass of *R. tingitana* [9,17,18,19]. Moreover, little is known about the bioactive composition of *R. tingitana*, and allelochemicals as well as the anti-dengue vector activity of *R. tingitana* extract was not previously studied. Despite its significance, little study has been conducted on the plant. In order to examine the biochemical components responsible for the biological effects, the current study used GC-MS spectroscopy to assess the chemical constituents of the Egyptian ecospecies of *R. tingitana* methanol extract. The study looked at the antioxidant activity of the extracted plant using the DPPH free radical scavenging assay, as well as the in vitro antibacterial activities against a range of pathogenic bacteria. The larvicidal activity of the extract against *A. aegypti* was also investigated.

## 2. Results and Discussion

### 2.1. Total Phenolic, Flavonoid, and Tannin Contents

The medicinal benefits of herbal medicines are based on the chemistry of plants. Knowing a plant’s chemical composition helps one comprehend its possible medicinal benefits. Numerous bioactive chemical substances known as secondary plant metabolites are created by the plant cell via the main metabolic pathways [20]. Among them, phenolic compounds were shown to have significant antioxidant [21], anti-inflammatory [22], antihyperglycemic [23], immunomodulatory, and anticancer [24] action. The present study demonstrated that *R. tingitana* shoot extract is filled with phenolics (143.68 mg gallic acid equivalent/g dry extract), flavonoids (32.41 mg catechin equivalent/g dry extract), and tannins (96.57 mg gallic acid equivalent/g dry extract). Recent studies include our own findings on the antioxidant properties and activities of phenolics and flavonoids. They both have the potential to substantially promote and inhibit oxidative processes in this regard. Additionally, it has been shown that they have the capacity to split up chains of free radicals, which has a strong antioxidant effect.. In addition to phenolics and flavonoids, tannins are a prevalent and significant plant phytochemical. Tannins have strong antibacterial [25] and antioxidant [21] properties. Tannins can have an antinutritional impact in addition to these outcomes [26]. Tannins are phenolic chemicals that exist naturally. They include a lot of phenolic groups and have a high molecular weight (Mr > 500), which leads to the precipitation of proteins [27].

Abdel-Mogib et al. [19] and Abd-ElGawad et al. [9] reported that aboveground biomass has undergone phytochemical analysis, which reveals the presence of phenolics, tannins, flavonoids, coumarins, volatile oils, glycosides, flavonoids, lactones, esters, significant levels of sterols, and/or triterpenes. Numerous intrinsic and extrinsic variables, specialized metabolic processes, and endogenous physiological changes in the plants all contribute to the variation in biochemical content in the various solvent extracts and within the plant parts [28].

### 2.2. Gas-Chromatography Mass Spectroscopy “GC-MS”

The chemical structures and constitutes of the *R. tingitana* extract were characterized by GC-MS analysis. Figure 1 demonstrates the relation between the relative abundance of the different components characterized by the extracted plant against the retention time at which a definite component was detected. The results in Table 1 confirmed that 17 components were interpreted from the extracted plant. Commonly, 6,10,14-trimethylpentadecan-2-one (21.98%) and methyl oleate (27.26%) are positioned as the major components, which were ascertained after 19.25, and 23.34 min, respectively. Subsequently, other constitutes were categorized with elevated percent of composition, for instance, palmitic acid (14.85%), (E)-octadec-9-enoic acid (2.16%), methyl palmitate (3.29%), (Z)-octadec-11-enoic acid (3.56%), 3,7,12-trihydroxycholan-24-oic acid (6.63%), and Stigmast-5-en-3-ol, (3α)- (7.57%) (Figure 2). The components of hydrocarbons were recorded at 13.56 “very low relative abundance” and 19.25 min, while the components of fatty acids and esters were recorded with retention times ranging from 16.62 to 33.50 min. The most abundant components of steroids were identified after retention time at 34.45–35.70 min.

The characterized components were classified under numerous naturally occurring categories, such as hydrocarbons (23.82%), fatty acids, esters of fatty acids (57.46%), steroids (17.26%), and terpenes (1.48%) (Figure 3). The class of fatty acids and/or their ester derivatives retained the majority of these components. Thus, two components were identified for hydrocarbons, ten components were characterized as fatty acid and esters, four components as steroids, and one component as a terpene. Accordingly, 6,10,14-trimethylpentadecan-2-one is the major constituent of the hydrocarbon category, with 21.98% of the total composition of the extracted components. Moreover, methyl oleate is the major constituent of the fatty acid esters class, with 27.26%, while Stigmast-5-en-3-ol, (3α)- as a steroidal sub-class is the major component under this category (7.57%).

### 2.3. Biological Characteristics of the Plant Extracts

#### 2.3.1. Antioxidant Activity—DPPH Assay

The capacity of *R. tingitana* methanolic extract to scavenge DPPH free radicals in comparison to ascorbic acid was used to determine its antioxidant activity. Half maximum inhibitory concentration (IC_50_) values were used to express the scavenging effects of plant extracts and the standard on the DPPH radical; the findings are shown in Table 2. A lower IC_50_ value indicates a greater capacity to scavenge DPPH radicals. Subsequently, the results, as presented in Table 2, confirmed that the extract from the shoot has the most antioxidant scavenging activity (IC_50_ = 30.77 mg L^−1^) compared to the ascorbic acid (IC_50_ = 12.02 mg L^−1^). The main element regulating the mechanism of the reactions engaged in the assessment of the antioxidant ability of the examined extract is the predominance of fatty acids and their derivatives (57.46%) and oxygenated hydrocarbons (23.82%) of total separated components. The present results of *Rumex vesicarius* were in agreement with the results of Abd-ElGawad et al. [9], Cornara et al. [29], and Leonti [30].

On the other hand, fatty acids and lipids, which were extracted from *Sisymbrium irio, Aesculus indica,* and *Abies pindrow*, showed strong antioxidant capabilities for scavenging the free radicals in the solution [31]. Palmitic acid and methyl oleate are the two main constituents, with 14.85 and 27.26% acting as significant antioxidant agents, respectively. The ability of reactive oxygen species, such as phenolics, fatty acids, terpenes, oxygenated hydrocarbons, or carbohydrates, to scavenge or stabilize free radicals generally determines the antioxidant capacity of bioactive chemicals [32,33,34].

In this investigation, the antioxidant activity of *R. tingitana* shoot extract was superior to that of other wild plant extracts in many countries [17]. According to Abd El-Gawad et al. [35], *R. tingitana* aerial parts have more antioxidant activity in their extract than other common Egyptian plants (*Cakile maritime, Centaurea glomerata, Juncus bufonius,* and *Lactuca serriola*). Different studies have demonstrated that the quantity of bioactive chemicals, particularly phenolic compounds, such as flavonoid, phenolic acids, ascorbic acid, and carotenoids, directly affects the antioxidant capabilities of plants [36]. From our study, this plant contains nonvolatile compounds, such as tannins, flavonoids and phenolics.

#### 2.3.2. Antibacterial Activity

The antimicrobial properties of methanolic extract of *R. tingitana* at a concentration of 10 mg L^−1^ against four Gram-negative and five Gram-positive bacterial strains have been assessed in this study using agar disc diffusion assay [21]. The results revealed that the methanolic extract of a certain plant effectively inhibits the growth of pathogenic bacteria with varying potency (Table 3). According to the current result, methanolic extract of *R. tingitana* had the maximum zone of inhibition against *S. typhimurium* and *B. cereus* (25.71 ± 1.63 and 24.42 ± 0.81 mm, respectively), while *C. tetani* and *S. xylosus* were the main resistant species (Table 3). The tested pathogenic organisms can be arranged in the following order based on their sensitivity: *S. typhimurium < B. cereus <*
*Escherichia coli < Staphylococcus aureus <*
*Klebsella pneumonius < Streptococcus pyogenes < Enterobacter aerogenes < S. xylosus <*
*C. tetani*. The common antibiotics, including ampicillin, azithromycin, cephradine, and tetracycline, had varying degrees of activity. *S. typhimurium* was totally resistant to ampicillin and cephradin, but *C. tetani* was completely resistant to cephradin and azithromycin and *E. aerogenes* was entirely resistant to tetracycline, ampicillin, and cephradin (Table 3). Additionally, plant extract demonstrated strong antibacterial activity against *B. cereus*, *E. coli*, and *S. typhimurium* (24.42 0.81, 22.85 0.87, and 25.71 1.63 mm, respectively) but less potency against *C. tetani* and *S. xylosus* when compared to antibiotics.

In the current investigation, the MeOH extract of *R. tingitana* had approximately twofold more activity against *S. aureus, S. xylosus*, and *E. coil* than did the chloroform extract in Yemen [37]. Additionally, the MeOH extract used in this investigation had stronger antibacterial action against *S. aureus* than that shown with MeOH derived from Tunisian ecospecies but was ineffective against *E. coil* [38]. Several previous experiments on the essential oil and n.hexane extract of the aerial parts of *R. tingitana* confirm that they were potent antibacterials against both Gram positive and negative bacteria [17]. These variations in the antibacterial potential may be explained by variances in the extracts’ chemical composition caused by environmental factors, such as geographical variability, soil type, and climate [39,40].

Generally, the Egyptian ecospecies showed more antimicrobial activity compared to the Yemen and Tunisian ecospecies. This difference could be ascribed to the variation in the chemical composition of the extract. The bioactivities of the plant extract are directly correlated with their chemical compound compositions either individually or synergistically [41]. Furthermore, it has been noted that MeOH extracts of wild plants contain a variety of different chemical compounds, including phenolic compounds and its derivative compounds, the esters of weak acids, fatty acids, and terpenes, and others. As a result, these chemical components can affect multiple target sites against the bacterial cells, with minor variations [40,42,43]. In order to determine the bacterial species’ efficacy against the extracts that were considered, its composition should be studied. Terpenes, steroids, oxygenated hydrocarbons, and carbohydrates showed stronger antibacterial potentials, according to our analysis of the main chemical constituents of the isolated *R. tingitana* [44,45,46]. Plant-derived phenolics, such as phenolic acids and flavonoids, two nonvolatile compounds found in this plant, have been shown to be effective antibacterial agents against bacteria with various resistances. Regardless of whether the bacteria are Gram-negative or Gram-positive species, the methanol extract of *R. tingitana* seemed to have stronger antibacterial power against a specific type of bacterial species.

#### 2.3.3. Anticancer Activity

The usage of therapeutic plants and the herbal extracts they are derived from, which are rich in polyphenolic chemicals, may help explain the decreased incidence of cancer. Numerous clinical trials have shown that different herbal medications can exhibit a range of anticancer effects. Additionally, the use of herbal extracts with distinctive therapeutic qualities is advantageous in the field of pharmaceutical chemistry due to their improved effectiveness and biological potency [47,48]. In this study, the cytotoxic activities of the prepared plant sample extract were evaluated using an MTT assay. The sample was tested in vitro against two tumor cells, i.e., HepG-2, and PC3 cell lines (Table 4). Doxorubicin was selected as a reference drug, comparing the results of the tested sample against the different cancer cells. The method was applied for determining the cell metabolic activity based on the aptitude of nicotinamide adenine dinucleotide phosphate (NADPH)-dependent cellular oxidoreductase enzymes to reduce the MTT tetrazolium dye to its formazan “insoluble” that has a purple color. The number of viable cells should increase with growth, decrease with cytotoxic treatments, and remain the same (or plateau) with cytostatic treatments. The control sample is a benefit for calculating the cell viability percent, as it produces 100% viability of healthy cells.

The experiments were run using five concentrations of each plant extract (1.56, 3.125, 6.25, 12.50, 25.00, 50.00, and 100 µg mL^−1^) prepared in a serial dilution. According to the results, the methanolic extracts of plant exhibited cytotoxic activity in a dose-dependent manner, which was comparable with doxorubicin as a reference standard. At 100 µg mL^−1^, the extract of *R. tingitana* showed inhibition activities of 93.14%, 91.41%, and 9.55% for HepG-2 and PC3 human tumor cells and normal cell (WI-38), respectively. However, the lowest concentration (1.56 µg mL^−1^) shows the lowest cytotoxic activity in all samples (Table 4).

The curves generated by plotting the percentages of cell survival vs. drug concentration (µM) were used to calculate the IC_50_ values, which expressed the concentration that represented 50% of the inhibition of cell growth. As a result, as the extract concentration and IC_50_ values decline, the potency of cytotoxicity will increase. For HepG-2 and PC3 cells, as well as WI-38 normal cells, the IC_50_ values of the MeOH extract from the *R. tingitana* sample were 29.977, 40.479, and >100 µg mL^−1^, respectively. To compare the outcomes of the tested compounds against the various cancer cells, which achieved IC_50_ values of 5.274, 8.303, and >100 µg mL^−1^ for HepG-2, PC3, and WI-38, respectively, doxorubicin was used as a reference drug (Table 4 and Figure 4). According to the IC_50_ data, the extracted shoot of this plant shows noncytotoxic activity for normal cells (WI-38) and moderate cytotoxic activity for two human tumor cells (HepG-2 and PC3). This result is in agreement with the result obtained by Csupor-Löffler et al. [49], who studied the ant proliferative activities of aqueous and organic extracts prepared from 26 Hungarian species of the family Asteraceae, which were tested in vitro against HeLa (cervix epithelial adenocarcinoma), A431 (skin epidermoid carcinoma), and MCF7 (breast epithelial adenocarcinoma) cells by using the MTT assay.

It is important to note that the structural nature of each extract’s components and the nature of the cancer cell line are frequently correlated with the cytotoxicity of extracted samples as cytotoxic agents on HepG2 and PC3 tumor cell lines. Furthermore, the characteristics of the extracted plant particles, such as their size, aggregation, and surface shape, may influence their cytotoxicity. The chemical composition of the plant extract, the kind of cancer cell, and the concentration of the extracted plant employed in this assessment were among the parameters that affected how effective the plant extract was as an anticancer agent for tumor cell proliferation [50]. Many polyphenols, such as isoflavones, are phytoestrogens that may bind to estrogen receptors and have an estrogenic impact on the tissues or organs they are intended to affect. Additionally, newly identified substances, such as phenolic acids and flavonoids, could be to blame for the anti-inflammatory and cytotoxic effects of plant extracts [40,51]. *R. tingitana* contains nonvolatile substances, such flavonoids and phenolics in the current investigation.

#### 2.3.4. Larvicidal Bioassay

Serious human diseases are spread by mosquitoes, which result in millions of fatalities each year. Due to the development of synthetic pesticide resistance, vector control is at risk. In the future, insecticides with a botanical origin could be a good substitute for biocontrol methods [52,53]. Data from Table 5 showed that *R. tingitana* methanolic extract had larvicidal action against *A. aegypti* larvae in their third instar at 24 and 48 h after treatment. Results from this investigation showed a substantial increase in larval mortality at all concentration of *R. tingitana* methanolic extract. The highest larval mortality was recorded at concentrations of 300 mg L^−1^ for 24, 48, and 72 h (32.54%, 44.28%, and 50.81%, respectively), while the lowest mortality was 6.35%, 10.13%, and 15.74% at 24, 48, and 72 h, when the concentration was found to be 100 mg L^−1^, compared to 1.05% for control group.

The LC_50_ (the concentration of extract that causes 50% death) of extract is 46.85, 35.75, and 29.38 mg L^−1^ for the various time periods of 24, 48, and 72 h, whereas the LC_90_ (the concentration of extract that causes 90% mortality) is 82.66, 63.82, and 53.30 mg L^−1^, respectively (Table 5). The different bioactive chemicals, such as phenolics, terpenoids, flavonoids, and alkaloids, either as single or combined compounds, determine the toxicity of the plants [40,54]. Additionally, desert plants, such as *R. tingitana*, are a rich source of bioactive chemicals that can be used as larvicides, pupicides, or mosquito repellents [55,56].

These outcomes also line up with those attained by Salama et al. [40], who utilized *Rumex vesicarius* methanolic extract against *A. aegypti* larvae and found that the LC_50_ and LC_90_ values were 19.99 and 36.12 mg L^−1^ after 24 h. Shehata [57] reported that petroleum ether extract from leaves of *Prunus domestica* and *Rhamnus cathartica* was more effective against *Culex pipiens* (LC_50_: 33.3 and 63.4 mg L^−1^, respectively) than chloroform (LC_50_: 70.8 and 192.1 mg L^−1^) and methanolic extracts (LC_50_: 132.7 and 273.5 mg L^−1^, respectively). Ullah et al. [56] recorded LC_50_ and LC_90_ values of Cassia fistula and Nicotiana tabacum extracts of 50.27, 203.99, and 17.77 and 206.49 mg L^−1^ against larvae of *Culex quinquefasciatus*. According to Dey et al. [58], *Piper longum* aqueous extract had the highest 24-h larval death rates when used to treat *A. aegypti*, *A. stephensi*, and *C. quinquefasciatus*.

## 3. Materials and Methods

### 3.1. Plant Material and Extraction Process

In April 2021, during the blossoming season, multiple populations of *Reichardia tingitana* (L.) Roth were gathered from in Al-Hashr Mountain, Jazan region, Saudi Arabia (17°27′21.31″ N 43°2′18.13″ E) (Figure 5). The plant was identified in line with Tackholm [4] and Boulos [6]. At Mansoura University in Egypt’s Faculty of Science, a voucher specimen (Mans. 0011820006) was made and added to the herbarium.

The samples were cleaned and allowed to dry naturally. A conical flask with a capacity of 250 mL and 150 mL of methanol was filled with 10 g of dried plant material. After that, the mixture was put into a water bath shaker (Memmert WB14: Schwabach, Germany), where it was shaken continuously for two hours at room temperature. The mixture was filtered using Whatman filter papers (no. 1, 125 mm, Cat. No. 1001 125, Germany). The methanol extract was dried using a rotary evaporator and the residue was collected in glass vials and stored in the refrigerator at 4 °C until further analyses [59].

### 3.2. Phytochemical Analysis

Similar to how hot infusions are used in conventional therapy, the active components were extracted using this technique. Twenty grams of the dried plant stems were mixed with 200 mL of deionized water and agitated for 30 min in a water bath system with a temperature of 70 °C. Filtering the extracted product, the filtrate was kept at 4 °C for subsequent use. The total quantities of phenolic, flavonoid, and tannin in the aqueous extract of *R. tingitana* aerial parts were quantified.

#### 3.2.1. Total Phenolic Contents

By using the Folin–Ciocalteu approach developed by Wolfe et al. [60] and Issa et al. [61], which used gallic acid as a standard, the total phenolic content was evaluated. Using a standard curve (y = 0.00621x, r^2^ = 0.976), the measured amounts of total phenolic content in the aqueous extract of *R. tingitana* were converted to milligrams of gallic acid/grams of dry plant extract.

#### 3.2.2. Total Flavonoid Contents

According to Zhishen et al. [62], who used catechin as a reference, a colorimetric analysis using aluminum chloride was used to determine the total flavonoids present. The amounts of total flavonoid constituents were calculated using a standard curve (y = 0.0027x, r^2^ = 0.979) as equivalents of catechin in milligram per dried plant extract in gram.

#### 3.2.3. Total Tannin Contents

The amount of total tannins was evaluated using the vanillin–hydrochloride test, and the values of the predicted samples were represented as the equivalents of tannic acid in grams/dried plant per 100 g, according to Burlingame [63] and Aberoumand [64].

### 3.3. Gas Chromatography-Mass Spectrometry Analysis (GC-MS)

The chemical composition of the extracted *R. tingitana* plant was described by employing the plant extract on a Trace GC-TSQ mass spectrometer (Thermo Scientific: Austin, TX, USA) with a direct capillary column TG-5MS (30 m × 0.25 mm × 0.25 m film thickness) [65]. The temperature of the column oven was first maintained at 50 °C, then raised by 5 °C per minute to reach 250 °C and maintained for 2 min, and then increased by 30 °C per minute to reach the final temperature of 300 °C and maintained for 2 min. The MS transfer line and injector were kept at temperatures of 260 and 270 °C, respectively. As a carrier inert gas, helium (He) was employed at a constant flow rate of 1 mL/min. After 4 min, the solvent was removed, and 1 µL diluted samples were immediately fed into the GC in split mode using an Auto sampler AS1300. In packed scan mode, EI mass spectroscopy data over the *m*/*z* range of 50–500 were gathered at an ionization voltage of 70 EV. The ion source’s temperature was set at 200 °C. By comparing the mass spectral data of the various extracted plant materials to those of the mass spectrometry databases WILEY 09 and NIST 14, it was possible to understand the chemical composition of each of the distinct plant materials. Five potential components were suggested by the GC-MS analysis for each identified peak. The probability factors and the primary structure’s fragmentation patterns governed the chosen structure in the case of the different suggested components.

### 3.4. Antioxidant DPPH Assay

The desired concentrations of *R. tingitana* were obtained by diluting a stock solution of the extracted plant in methanol (5, 10, 20, 30, 40, and 50 mg L^−1^). To each concentration of the sample solution that had been made, DPPH solution (1 mL, 0.135 Mm) was added. The tested samples’ concentrations of catechol were utilized as the standard. The samples were left at room temperature in the dark for 30 min, and a UV/Vis spectrophotometer was used to measure the absorbance at a wavelength of 517 nm (Spekol 11 spectrophotometer, analytic Jena AG: Jena, Germany). The following equation was used to compute the percentages of antioxidant scavenging activities, using a DPPH solution in methanol as a reference:(1)$$\mathrm{Inhibition}\;\left(\%\right)=\frac{A\;\mathrm{control}\;-\;A\;\mathrm{sample}}{A\;\mathrm{control}}\times 100$$

The approach was applied while making only minimal alterations to the preceding investigations [66,67]. The inhibitive concentrations were calculated using the exhibited exponential curve [68], which depicted the connection between the sample concentration and the amount of residual DPPH^•^ radical (IC_50_, mg L^−1^).

### 3.5. Antimicrobial Activity Procedure

The agar well diffusion experiment was used to determine the antibacterial potential of the examined *R. tingitana* extract [69,70]. Standard antibiotics were cephradin (CPD), tetracycline (TAC), azithromycin (ATM), and ampicillin (AMC). The microbial species were purchased from the Microbiological Resources Centre (MIRCEN), Faculty of Agriculture, Ain Shams University, and were of animal origin. The tested bacteria were *Escherichia coli* (NR_112558.1), *Enterobacter aerogenes* (KX878983.1), *Salmonella typhimurium* (NR_074910.1), and *Klebsella pneumonius* (NR_117683.1). Gram-positive bacteria: *Bacillus cereus* (EMCC_1080), *Clostridium tetani* (KHO40477.1), *Staphylococcus aureus* (NR_115606.1), *Streptococcus pyogenes* (MH916557.1), and *Staphylococcus xylosus* (NR_113350.1). The inhibitory zone diameters (mm) were determined after the plates had been incubated at 37 °C for 18 to 24 h.

### 3.6. Anticancer Activity Procedure

Hepatocellular carcinoma (HePG-2), mammary gland carcinoma (MCF-7), and human prostate cancer (PC3) are three particular human tumor cell lines that were purchased from the ATCC holding business for biological products and vaccines (VACSERA: Cairo, Egypt). A common chemotherapeutic anticancer medication was doxorubicin. The chemical reagents used were RPMI-1640 medium, MTT, DMSO (Sigma Co.: St. Louis, MO, USA), and fetal bovine serum (FBS; Gibco Life Technologies: Paisley, UK).

A typical colorimetric MTT assay was performed to monitor cell growth and assess the cytotoxicity of the extracted *R. tingitana* in accordance with the guidelines given by Bondock et al. [71]. Simply put, the transformation of MTT (2-(4,5-dimethylthiazol2-yl)-3,5-diphenyl-2H-tetrazolium bromide) from yellow to purple was carried out by mitochondrial succinate dehydrogenases of living cells. To create the cell strains, RPMI-1640 media was supplemented with 10% fetal bovine serum. In an incubator with 5% CO_2_, penicillin (100 units mL^−1^) and streptomycin (100 g mL^−1^) antibiotics were added. Cell lines were seeded on a 96-well plate at a density of 1.0 × 10^4^ cells/well suspended in 100 μL of complete medium and incubated at 37 °C for 48 h with 5 % CO_2_. Cells were cultured for the first 24 h, and then treated with various doses of the test samples. MTT solution (5 mg mL^−1^, 20 L) was added following a 24-h drug treatment period, and the mixture was once more incubated for 4 h. To dissolve the created violet formazan, 100 L of DMSO was then applied to each well. The colorimetric analysis was conducted, and a plate reader was used to measure the absorbance values at 570 nm (EXL 800, New York, NY, USA). The Origin 8.0^®^ program (OriginLab Corporation, Available online: https://www.originlab.com/; Accessed 27 March 2021) was used to generate the IC_50_ values using nonlinear regression (sigmoid type). The following equation, in which optical density (OD) expresses the absorbance read of the control and the tested sample, was used to compute the percentage of inhibition in cell growth:(2)$$\%\;\mathrm{Inhibition}=\frac{\mathrm{OD}\;\mathrm{control}\;–\mathrm{OD}\;\mathrm{sample}}{\mathrm{OD}\;\mathrm{control}}\times 100$$

### 3.7. Mosquitocidal Assay

#### 3.7.1. Rearing of *Aedes aegypti*

The Center for Disease Vector in Jizan provided the *A. aegypti* larvae, which were raised for six generations in the Center for Environmental Research at the Faculty of Science in Jazan University under carefully controlled conditions of temperature (28 ± 2 °C), relative humidity (70–80%), and 12h/12h light–dark regime. Three days after emergence, adult mosquitoes were housed in (30 × 30 × 30 cm) wooden cages and given cotton pieces every day that had been soaked in a 10.0% sucrose solution. Following this time, females were permitted to have a blood meal from a pigeon host, which is required for egg laying (anautogeny). In the cage for egg laying, a plastic cup oviposition (15 × 15 cm) with dechlorinated tap water was inserted. The finished egg rafts were removed from the plastic dish and placed in plastic pans (25 × 30 × 15 cm) with three liters of tap water that had been left for 24 h. A slice of bread was given to the developing larvae each day as food. This diet was discovered to be the best for female fertility and larval growth [72].

#### 3.7.2. Larvicidal Activity Procedure

The tested material for larvicidal activity was dissolved in 0.1 mL of methanol. To find deaths, several concentration ranges of each relevant extract were generated. In 200 mL plastic cups containing 100 mL of dechlorinated tap water, the tested sample was run. Then, 10 larvae in the third instar were immediately placed into plastic containers that contained various extract concentrations. Typically, three replicates were employed for each concentration under examination. All of the plastic cups were incubated in a mosquito colony under controlled conditions and, thereafter, mortality was noted. A total of 0.1 mL of methanol or 2 drops of Tween 80 were given to control larvae in 100 mL of water. Up to adult emergence, mortality was tracked every day and dead larvae and pupae were eliminated [72]. Failure to react to mechanical stimulation was used to detect larval death. The following equation [73] was utilized to calculate the % larval mortality:(3)$$\mathrm{Larval}\;\mathrm{mortality}\;\left(\%\right)=\frac{\mathrm{Number}\;\mathrm{of}\;\mathrm{dead}\;\mathrm{larvae}}{\mathrm{Number}\;\mathrm{of}\;\mathrm{treated}\;\mathrm{larvae}}\times 100$$

### 3.8. Statistical Analysis

Using the Costat software (CoHort Software: Monterey, CA, USA), the antioxidant, antibacterial, and larvicidal activity experiments were conducted three times with three replications. The outcomes were then subjected to a one-way ANOVA to assess the significance of the differences between samples.

## 4. Conclusions

In conclusion, by using GC-MS analysis, 17 components were interpreted from the MeOH extract of *R. tingitana* aerial parts. The two primary components are commonly reported as 6,10,14-trimethylpentadecan-2-one (21.98%) and methyl oleate (27.26%). Hydrocarbons (23.82%), fatty acids, esters of fatty acids (57.46%), steroids (17.26%), and terpenes (1.48%) were identified as the major components. The *R. tingitana* shoots demonstrated advantageous biological characteristics, including larvicidal effectiveness against *A. aegypti*, the dengue virus vector, as well as antioxidant, antibacterial, and cytotoxic agents. Shoot extract exhibited the highest antioxidant activity, showing a better capacity to trap free radicals in the DPPH solution, with an IC_50_ value of 30.77 mg L^−1^. The current findings indicate that the *R. tingitana* extracted shoot exhibited the most potent antibacterial activity against Gram-negative bacterial species, exhibiting broad-spectrum activity against *S. typhimurium* and *B. cereus*, while *C. tetani* and *S. xylosus* were the main resistant species. Additionally, compared to control, the methanol extract of *R. tingitana* showed increased potential anticancer activity, with significant cytotoxicity for the two tumor cells: HepG-2 and PC3 cells. Furthermore, *A. aegypti* larvae were more easily killed by the MeOH extract of the plant sample. The significant biological findings and the impressive percentage of phenolic, flavonoid, and tannin contents found in the *R. tingitana* extract, supported the possibility of further study on this plant for the creation of drugs from natural sources.

## Figures and Tables

**Figure 1 plants-11-02028-f001:**
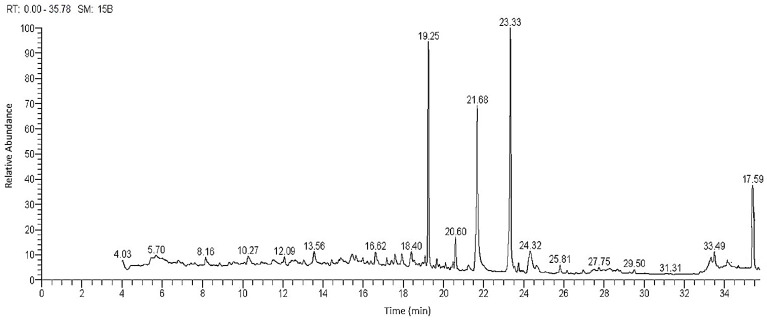
Chromatogram and structures of main components of the methanol extract of *R. tingitana* shoot by GC-MS.

**Figure 2 plants-11-02028-f002:**
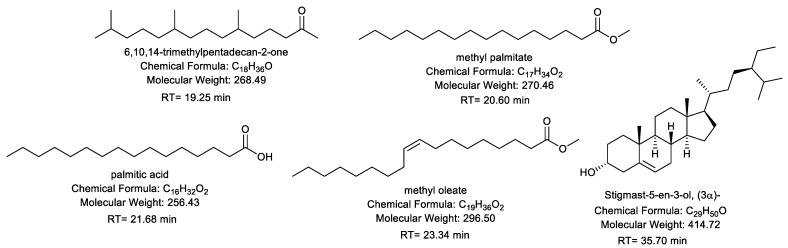
Chemical structure of the major identified compounds in the MeOH extract of *R. tingitana*.

**Figure 3 plants-11-02028-f003:**
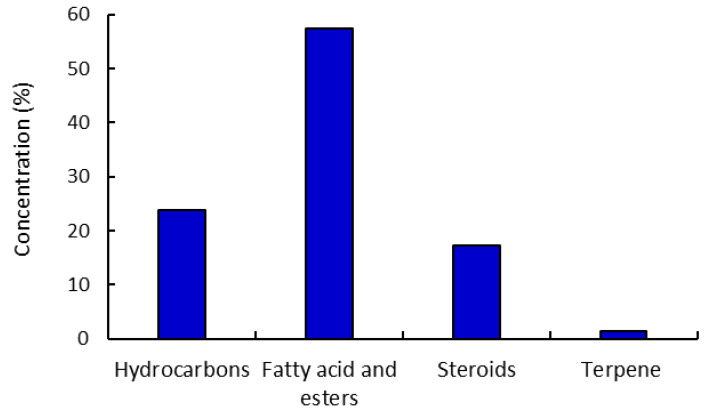
The categorized chemical constitutes identified from the extracted *R. tingitana* by GC-MS analysis.

**Figure 4 plants-11-02028-f004:**
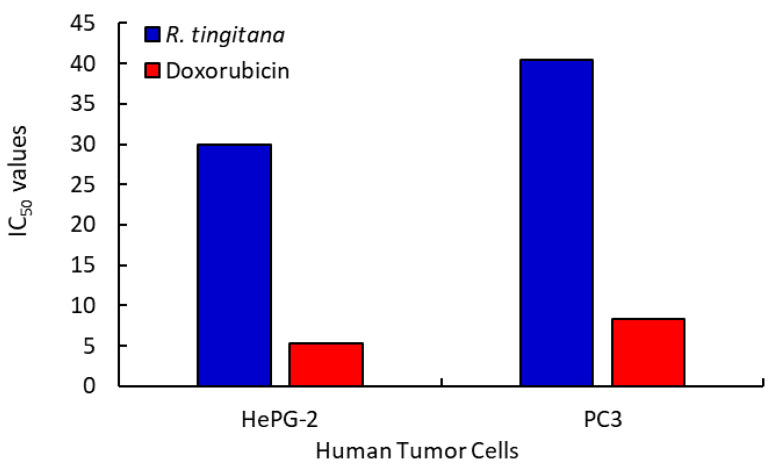
IC_50_ values of the tested plant sample and doxorubicin as standard against human cancer cells.

**Figure 5 plants-11-02028-f005:**
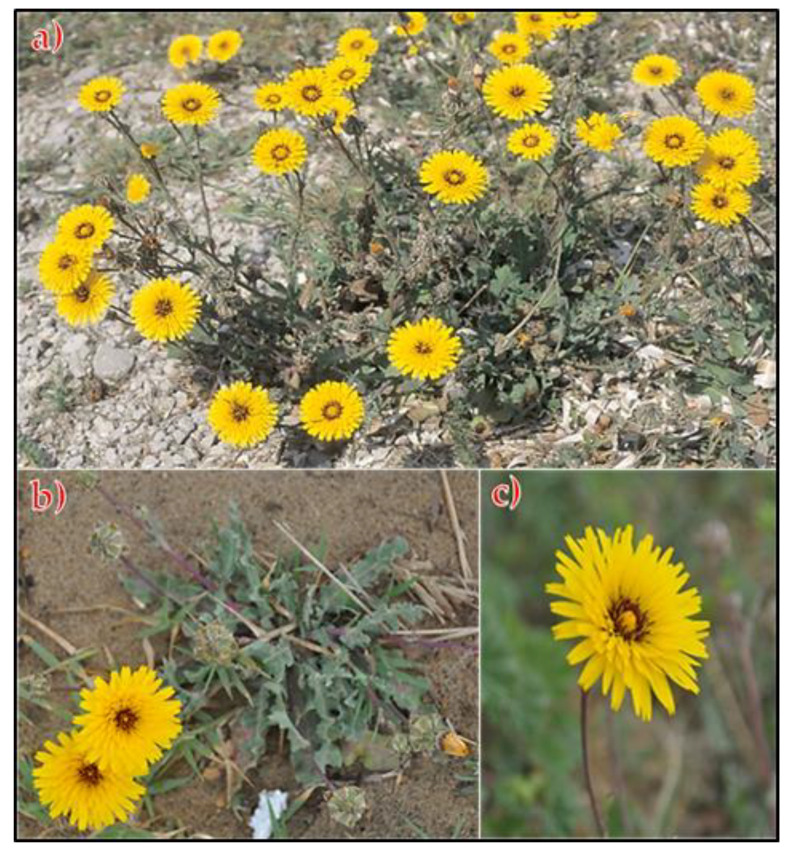
*Reichardia tingitana* (L.) Roth plant. (**a**) Overview of the growing herb, (**b**,**c**) close view of vegetative stage and flowering branch.

**Table 1 plants-11-02028-t001:** The characterized chemical components were isolated from the extracted shoots of *R. tingitana*.

No.	RT ^a^	Conc. % ^b^	Compound	Molecular Weight	Molecular Formula
Hydrocarbons					
1	13.56	1.84 ± 0.04	5,5,8a-trimethylhexahydro-2H-chromen-4a(5H)-yl acetate	240.34	C_14_H_24_O_3_
2	19.25	21.98 ± 0.22	6,10,14-trimethylpentadecan-2-one	268.49	C_18_H_36_O
Fatty Acid and Esters					
3	16.62	2.16 ± 0.05	(E)-octadec-9-enoic acid	282.47	C_18_H_34_O_2_
4	17.92	1.44 ± 0.03	Ethyl (9Z,12Z)-octadeca-9,12-dienoate	308.51	C_20_H_36_O_2_
5	19.08	1.13 ± 0.02	Oleic acid	282.47	C_18_H_34_O_2_
6	19.67	1.16 ± 0.02	Isobutyl octadecyl phthalate	474.73	C_30_H_50_O_4_
7	20.6	3.29 ± 0.07	Methyl palmitate	270.46	C_17_H_34_O_2_
8	21.68	14.85 ± 0.18	Palmitic acid	256.43	C_16_H_32_O_2_
9	23.34	27.26 ± 0.24	Methyl oleate	296.5	C_19_H_36_O_2_
10	23.74	1.06 ± 0.02	Methyl stearate	298.51	C_19_H_38_O_2_
11	24.31	3.56 ± 0.06	(Z)-octadec-11-enoic acid	282.47	C_18_H_34_O_2_
12	33.5	1.55 ± 0.03	2-hydroxypropane-1,3-diyl (9E,9′E)-bis(octadec-9-enoate)	621	C_39_H_72_O_5_
Steroids					
13	18.4	1.75 ± 0.04	Estra-1,3,5(10)-trien-17α-ol	256.39	C_18_H_24_O
14	34.45	1.31 ± 0.02	Ethyl 3,7,12-trihydroxycholan-24-oate	436.63	C_26_H_44_O_5_
15	35.39	6.63 ± 0.05	3,7,12-trihydroxycholan-24-oic acid	408.58	C_24_H_40_O_5_
16	35.7	7.57 ± 0.08	Stigmast-5-en-3-ol, (3α)-	414.72	C_29_H_50_O
Terpene					
17	17.59	1.48 ± 0.02	Corymbolone (4a-hydroxy-4,8a-dimethyl-6-(prop-1-en-2-yl) octahydronaphthalen-1(2H)-one)	236.36	C_15_H_24_O_2_
Total		100.0			

^a^ retention time, ^b^ average concentration of three replications ± standard deviation.

**Table 2 plants-11-02028-t002:** Radical scavenging activity percent (%), and IC_50_ values (mg L^−1^) at various concentrations of the methanol extracted of *R. tingitana* and the standard ascorbic acid by DPPH assay.

Treatment	Conc. (mg L^−1^)	Radical Scavenging Activity (%)	IC_50_ (mg L^−1^)
*R. tingitana*	5	8.32 ± 0.42 ^F^	30.77
	10	23.84 ± 1.64 ^E^	
	20	42.27 ± 2.37 ^D^	
	30	51.61 ± 2.88 ^C^	
	40	62.33 ± 3.20 ^B^	
	50	71.91 ± 3.72 ^A^	
	LSD_0.05_	1.62 ***	
Ascorbic acid	1	2.81 ± 0.01 ^F^	12.02
	2.5	11.38 ± 0.03 ^E^	
	5	38.57 ± 0.19 ^D^	
	10	47.92 ± 0.51 ^C^	
	15	61.34 ± 1.42 ^B^	
	20	72.61 ± 1.55 ^A^	
	LSD_0.05_	1.40 ***	

Values are average (n = 3) ± standard deviation. LSD_0.05_ expressed the calculated least of the smallest significance between two means, as each test was run on those two means (calculated by Factorial ANOVA). Different superscript letters within each treatment (column) express significant variation at a probability level of 0.05 (Duncan’s test). ***: significant at *p* ≤ 0.001.

**Table 3 plants-11-02028-t003:** Antibacterial activity of the methanol extract from the aerial parts of *R. tingitana* and some selected reference antibiotics at a concentration of 10 mg mL^−1^.

Microbes	*R. tingitana* (10 mg mL^−1^)	Standard Antibiotic (10 mg L^−1^)			
		Cephradin	Tetracycline	Azithromycin	Ampicillin
Gram-Negative Bacteria					
*Escherichia coli*	22.85 ± 0.87 ^B^	17.81 ± 0.82 ^D^	20.34 ± 0.71 ^BC^	20.45 ± 0.51 ^B^	20.51 ± 0.73 ^C^
*Enterobacter aerogenes*	11.36 ± 0.44 ^E^	0.00 ^F^	0.00 ^F^	14.56 ± 0.63 ^C^	0.00 ^F^
*Salmonella typhimurium*	25.71 ± 1.63 ^A^	0.00 ^F^	12.82 ± 0.54 ^D^	9.35 ± 0.07 ^D^	0.00 ^F^
*Klebsella pneumonius*	15.13 ± 0.51 ^D^	12.08 ± 0.42 ^E^	20.17 ± 0.68 ^C^	13.75 ± 0.49 ^C^	8.03 ± 0.09 ^E^
Gram-Positive bacteria					
*Bacillus cereus*	24.42 ± 0.81 ^AB^	20.33 ± 0.55 ^BC^	12.09 ± 0.50 ^D^	19.44 ± 0.42 ^B^	10.62 ± 0.42 ^D^
*Clostridium tetani*	9.25 ± 0.04 ^F^	0.00 ^F^	8.54 ± 0.05 ^E^	0.00 ^D^	10.51 ± 0.27 ^D^
*Staphylococcus aureus*	18.62 ± 0.64 ^C^	21.82 ± 0.57 ^B^	22.77 ± 0.61 ^AB^	20.08 ± 0.91 ^B^	30.67 ± 1.92 ^A^
*Streptococcus pyogenes*	13.66 ± 0.32 ^D^	25.60 ± 1.31 ^A^	23.63 ± 1.58 ^A^	23.74 ± 0.87 ^A^	22.07 ± 1.37 ^C^
*Staphylococcus xylosus*	10.54 ± 0.51 ^E^	19.73 ± 0.98 ^C^	21.15 ± 1.65 ^ABC^	22.81 ± 0.69 ^A^	25.32 ± 1.51 ^B^
LSD_0.05_	0.0001 ***	0.0001 ***	0.0001 ***	0.0001 ***	0.0001 ***

Value is the diameter of the inhibition zone (mm) as an average of three replications ± standard error. Different superscript letters within each treatment (column) express significant variation at a probability level of 0.05 (Duncan’s test). LSD: least significant difference. *** *p* < 0.001.

**Table 4 plants-11-02028-t004:** Cytotoxic activity and the IC_50_ values of the *R. tingitana* MeOH extract against the tumor and normal cells at different concentrations, and doxorubicin as standard. Hepatocellular carcinoma (HePG-2), human prostate cancer (PC3), and normal cell (WI-38).

Samples	Conc. (µg mL^−1^)	In Vitro Cytotoxicity (%)		
		HePG-2	PC3	WI-38
*R. tingitana*	100	93.14	91.41	9.55
	50	88.34	81.62	8.33
	25	85.07	75.31	5.58
	12.5	68.52	58.22	3.67
	6.25	50.59	41.72	1.48
	3.125	38.99	25.38	1.32
	1.56	25.62	22.93	0.95
	IC_50_	29.977	40.479	>100
Doxorubicin	100	63.15	70.07	12.23
	50	54.03	46.83	9.52
	25	49.58	37.61	6.31
	12.5	41.39	32.42	4.98
	6.25	34.56	24.27	3.32
	3.125	23.14	14.65	2.45
	1.56	6.47	3.87	1.13
	IC_50_	5.274	8.303	>100

IC_50_: inhibitory concentration (µg): 1–10 (very strong), 11–20 (strong), 21–50 (moderate), 51–100 (weak), and above 100 (noncytotoxic).

**Table 5 plants-11-02028-t005:** Larvicidal effect of the methanol extract from the aerial parts of *R. tingitana* on 3rd larval instar of *A. aegypti*.

Conc. (mg L^−1^)	*R. tingitana*		
	24 h Post-Treatment	48 h Post-Treatment	72 h Post-Treatment
300	32.54 ± 1.41 ^A^	41.28 ± 2.03 ^A^	50.81 ± 2.37 ^A^
250	28.33 ± 1.23 ^A^	31.51 ± 1.39 ^B^	42.17 ± 1.83 ^AB^
200	18.16 ± 0.79 ^B^	29.37 ± 1.25 ^B^	34.62 ± 1.44 ^BC^
150	11.32 ± 0.42 ^C^	19.09 ± 0.83 ^C^	28.06 ± 1.07 ^C^
100	6.35 ± 0.31 ^CD^	10.13 ± 0.44 ^CD^	15.74 ± 0.52 ^D^
Control	1.05 ± 0.05 ^D^	1.05 ± 0.05 ^D^	1.05 ± 0.05 ^E^
F-value	166.13 ***	110.75 ***	142.10 ***
*p*-value	<0.0001 ***	<0.0001 ***	<0.0001 ***
LC_50_	46.85	35.75	29.38
LC_90_	82.66	63.82	53.30

Mortality was expressed as mean ± SE (standard error) of 3 replicates. LC_50_: median lethal concentration; LC_90_: lethal concentration. Different superscript letters within each treatment (column) express significant variation; *** all F-values are significant at *p* ≤ 0.001 (Duncan’s test).

## Data Availability

The data presented in this study are available on request from the corresponding author.

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
