# Peer review of "Chemical Composition of Reichardia tingitana Methanolic Extract and Its Potential Antioxidant, Antimicrobial, Cytotoxic and Larvicidal Activity"

_plants, 2022, doi:10.3390/plants11152028_

Round 1

Reviewer 1 Report

 In general, the work seems good, where appropriate methods have been used. However, here are some suggestions to further improve the manuscript before publication:

1- Write authority of the test plant along with the scientific name at first citation in the Abstract.

2- While writing scientific names of the organisms, write full scientific name at the first citation in the text, thereafter, can abbreviate the genus name (only the first letter of genus name) throughout the paper. 

3- After writing full name of the Aedes aegypti in Italics at first citation, write A. aegypti instead of Ae. aegypti throughout the text.

4- Give proper spaces between the digits and its units.

5- In Introduction, a few statements are without references. I have suggested a few latest and relevant references that the author can cite after checking their suitability and relevance.

Reviewer 2 Report

Thank you so much for this interesting work. Even so, I have a big concern…

Why do the authors use methanolic extracts knowing that this is toxic for humans?

Line 17: “around 100%”

Lines 32 and 33: Reformulate the last sentence, please.

Line 37: “they are rich in”

Lines 86-88: “Among them, phenolic compounds were shown to have significant antioxidant [19], anti-inflammatory [20], 87 antihyperglycemic [21], immunomodulatory, and anticancer [22] action”

Line 90: insert spaces “143.68 mg gallic acid equivalent/g dry extract), flavonoids (32.41 mg”

Lines 97-98: Please, reformulate this sentence.

Line 100: “which leads to the precipitation of proteins”

Section 2.2.1. Please, compare with the IC50 value of ascorbic acid.

Table 2 column 2, something is not right, because we need to compare the same concentrations of the extract and the control. The IC50 value is ok for both and make sense, but the previous column, no, because the authors tested different concentrations.

Make correlations between phytochemical amounts and the biologic potential

Line 184: Please, insert the citations and deeper this topic.

Is it impossible to perform other antioxidant assays, like against superoxide and nitric oxide since, contrary to DPPH, these radials are present in human body?

Line 237, and 301-302: Flavonoids are phenolics, this sentence needs to be reformulated. Correct for “flavonoids” not “flavonides”

Line 347: use the Celsius degree symbol.

What are the criteria of the authors to collect the samples?

Line 361: “Twenty”

Mention cells passage.

Please, we need to reformulate the article, in order to the discussion be in the same order of material and methods.
